# KV-DISTILL: NEARLY LOSSLESS CONTEXT COMPRESSION FOR TRANSFORMERS

## ABSTRACT

Sequence-to-sequence natural language tasks often benefit greatly from long contexts, but the quadratic complexity of self-attention in standard Transformers renders usage of long contexts non-trivial. In particular, during generation, temporary representations (stored in the so-called KV cache) account for a large portion of GPU memory usage, and scale linearly with context length. In this work, we introduce KV-DISTILL , a flexible compression framework for large language models (LLMs) that distills long context KV caches into significantly shorter representations. KV-DISTILL can be trained as a parameter-efficient adaptor for pre-trained models, and enables the compression of arbitrary spans of a context while preserving the pre-trained model's capabilities, including instruction-tuning. We do this by treating a compressed-uncompressed cache as a student-teacher pairing and applying a KL-type divergence to match the generated outputs. Our experiments show that KV-DISTILL outperforms other compression techniques in worst-case extractive tasks, and approaches uncompressed performance in long context question answering and summarization. Furthermore, KV-DISTILL can be fine-tuned on domain-specific contexts to reduce context lengths by up 95% while preserving downstream task performance. We demonstrate the generalizability of KV-DISTILL across various model sizes and architectures. Our code and checkpoints will be made available at https://example.com

## 1 INTRODUCTION

Harnessing the full potential of attention-based large language models (LLMs) often requires them to condition on long contexts. However, use of expansive contexts is complicated by the quadratic complexity of self-attention. In particular, during generation, one must maintain a store of all past key and value representations of past tokens (called the KV cache) that grows linearly with sequence length. The memory burden imposed by the KV cache is significant, and often limits the length of the sequences that a model can handle.

Much work has been devoted to architectural improvements to the attention mechanism, with the aim of reducing the aforementioned memory burden during generation. Strategies include augmenting sequences with memory tokens (Rae et al., 2020; Wu et al., 2022), sparsifying attention patterns (Beltagy et al., 2020), and using conditional computation to only process essential tokens (Ainslie et al., 2023). However, such techniques have seen little widespread adoption due to performance drops on downstream tasks, or inefficient training/inference procedures. Furthermore, even when given long contexts, recent work has shown that LLMs fail to fully utilize them (Qin et al., 2022; Liu et al., 2024; Lu et al., 2024). Taken together, this suggests that long contexts may allow for significant compression while yielding large memory savings.

Prior work in parameter-free context compression has primarily focused on how to select representations in the KV cache for eviction, with promising results (Zhang et al., 2023). However this can suffer large performance drops under high compression ratios. Furthermore, we suppose that there is room for further performance improvements in general-purpose context compression when the model is trained to account for compression. Earlier studies in this area have typically utilized a combination of cross-entropy and autoencoding objectives to pre-train general context compressors (Qin et al., 2024; Ge et al., 2024; Rae et al., 2020). These loss functions have led to signficant performance loss at high compression rates and fail to maintain the carefully-tuned instruction-following

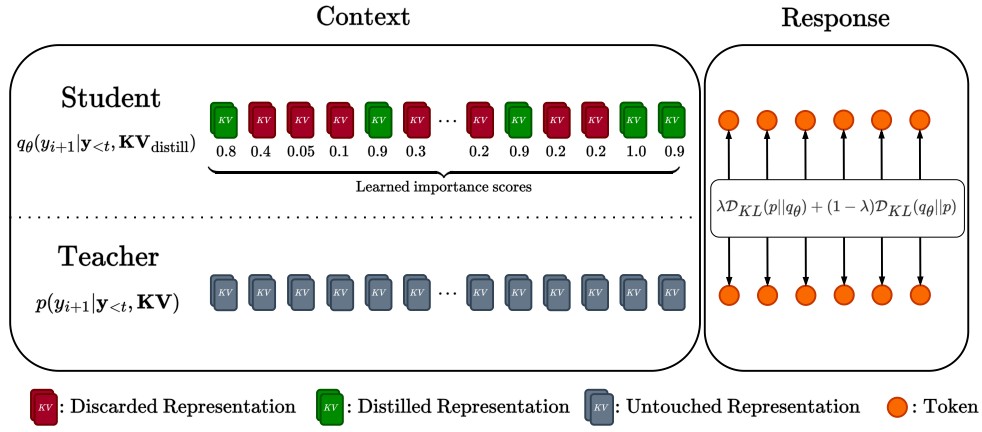

Figure 1: We subselect tokens from the KV cache and distill into the smaller subset

capabilities of modern language models, or otheriwse require additional instruction-tuning training to maintain their capabilities. In addition, these works frequently rely on the BLEU metric (Papineni et al., 2002) to evaluate the compressive capability of trained models, and devote less attention to assessing how models' actually use the compressed cache. Furthermore, they suffer from large drops in performance under high compression ratios, oddly underperforming parameter-free methods.

In this work, we design a general-purpose trainable context compression method for LLMs that outperforms prior methods. Our method, KV-DISTILL, retains pretrained model capabilities, is suitable for long contexts, and has minimal performance penalty on downstream tasks, while supporting coherent, useful generation at compression ratios as high as 1000x.

To achieve this we train a scorer which retains the most important context tokens, while applying a parameter efficient adapter to conditionally modify important tokens' activations in-place. We further apply a token-level KL-type divergence to match the next-token prediction distributions, treating the compressed cache as a student, and the uncompressed cache as a teacher. KV-DISTILL only need be applied once to a fixed context, has zero overhead during auto-regressive decoding, and can compress arbitrary (sub)spans of a given context. We show improvements on several model families, considering extractive and abstractive tasks, with both short and long contexts, and at multiple model scales. KV-DISTILL is general purpose and has broad applicability to the LLM community.

## 2 BACKGROUND

### 2.1 KEY-VALUE CACHE

Transformer-based language models (LMs)(Vaswani et al., 2017) use self-attention to aggregate context information and make predictions. A decoder-only transformer LM *autoregressively* predicts new tokens, and each step requires the LM to obtain the key and value states of all past tokens. To avoid re-computing the KV state of past tokens, most LM implementations (e.g. Wolf et al. (2020)) cache the key and values states, in a structure called the KV cache. When making new predictions, self-attention is performed on query states of the new token and the KV -cache, and the new token's key and value representations are appended to the KV cache.

Because the KV cache grows proportional to the number of tokens generated, maintaining the full KV cache in memory is a primary bottleneck when conditioning on large contexts. The goal of this work is to alleviate this by *compressing KV cache in the dimension of sequence length*.

### 2.2 RELATED WORK

Much prior work has tackled the problem of reducing the complexity of the self-attention mechanism itself. Previous work tries to sparsify the attention patterns(Beltagy et al., 2020; Zaheer et al., 2020),

use recurrence attention(Yang et al., 2019), or kernelize the attention matrix(Choromanski et al., 2021), but they require a considerable amount of further training. Another research direction tries to extrapolate the existing LMs to long context without heavily re-training them. For example, Bertsch et al. (2023); Tworkowski et al. (2023) use techniques akin to $k$NN to directly extend the attention range of LLMs without any further training.

Similar to our work, another line of work involves compressing the hidden states (KV cache) of past tokens into a shorter sequence of representations. For example, some work learns "soft represenatation" of context(Qin & Eisner, 2021). Mu et al. (2023) compress particular prompts into much shorter "gist tokens", but do not attempt more general context compression. Furthermore, their method demonstrates poor generalizability, as performance does not scale with the number of gist tokens used. Zeng et al. (2023) propose to recognize and prioritize some important tokens (VIP tokens) during inference. More relevant to our experiments, the following three methods employ a similar idea of dynamically compressing the context prior to inference:

Ge et al. (2024) design In-Context Autoencoder (ICAE) to compress long contexts for use in large language models (LLMs). ICAE consists of two main components: a learnable encoder and a fixed decoder. The encoder compresses the input context into a small number of memory slots. These memory slots are then used by the frozen LLMs (decoder) to reconstruct the context or respond to prompts. ICAE is pretrained using autoencoding and language modeling objectives on a large pre-training corpus and further fine-tuned using instruction data to maintain instruction-tuning. However, there is still a gap in downstream task performance when using an ICAE-compressed context, compared to an uncompressed context, and the method falters under high compression ratios.

Qin et al. (2024) propose DODO to compress sub-select KV activations to a set of "nugget" tokens, which grow proportionally with the length of context sequence. Their method is trained with auto-encoding or language modeling objectives. However, DODO models operate at a fixed compression ratio, require training both an encoder and decoder, and still show a large gap in downstream task performance when compared to an uncompressed context.

Zhang et al. (2023) propose $H_2O$ to reduce memory usage during generation. The technique identifies "heavy-hitter" tokens, which significantly influence attention scores during inference. These tokens are retained while less important ones are evicted in a greedy fashion. Although nearing uncompressed performance, the authors note that $H_2O$ performance falls at compression ratios over 20x. More importantly, $H_2O$ offers no way to further improve compressive performance given prior domain knowledge.

## 3 KEY-VALUE DISTILLATION

### 3.1 OVERALL PROCEDURE

In this paper, we consider a transformer-based language model (Vaswani et al., 2017), denoted by LM, that is defined on the vocabulary $\mathcal{V}$. The overall procedure of KV-DISTILL is as follows. First, a set of important tokens in the input context is determined. Next, we use an adapted language model $LM_\theta$ to encode the context into a KV cache, and sub-select the important tokens from the generated KV cache. Finally, the unmodified LM conditions on the compressed KV cache to auto-regressively generate it's output.

### 3.2 HIDDEN STATE SUBSELECTION FOR KV CACHE COMPRESSION

Let $\mathbf{c} = \{w_i\}_{i=1}^N$ represent a context consisting of $N$ tokens, where $w_i \in \mathcal{V}$ and $\mathbf{c} \in \mathcal{V}^N$. In a typical scenario, LM predicts a sequence of new tokens, denoted by $\mathbf{y}$, conditioning on $\mathbf{c}$. For example, $\mathbf{c}$ may be a prompt and LM generates $\mathbf{y}$ as a response. Future token prediction draws on information from past tokens via *attention* by having LM encode the context tokens into key and value hidden states $\mathbf{X}_l^{(K)}, \mathbf{X}_l^{(V)} \in \mathbb{R}^{N \times d}$, which taken together form the KV cache (Section 2.1), where $d$ is the dimension of the transformer and $l$ is the layer of LM. In the remainder of this paper, we may drop the subscript $l$ and superscripts $^{(K)}$ and $^{(V)}$ and use $\mathbf{X}$ to generally denote the key/value states of transformers at any layer.

Transformers assume that $\mathbf{X}$ fully describes and represents the context $\mathbf{c}$. However, attending to $\mathbf{X}$ can be inefficient when $\mathbf{c}$ is long. Therefore, we further assume that retaining *a subset of key/value states is sufficient for approximating the next-token distribution conditioned on all key/value states.* That is, we could *retain* rows from $\mathbf{X}$ to form $\tilde{\mathbf{X}} \in \mathbb{R}^{k \times d}$, where $k \leq N$ is the number of selected rows. We use a subset of the tokens' hidden states to represent the complete context, which is plausible because representations in $\tilde{\mathbf{X}}$ are conditioned on the prior context. Formally, suppose we determine the $(i_1, \ldots, i_k)$-th tokens are to be retained in layer $l$. We use a hard selection matrix $\mathbf{S}_l \in \{0, 1\}^{k \times N}$ to derive (layer-specific) $\tilde{\mathbf{X}}_l$ from $\mathbf{X}_l$ by

$$\tilde{\mathbf{X}}_l = \mathbf{S}_l \mathbf{X}_l, \quad \mathbf{S}_l = [\mathbf{e}_{(i_1)}, \ldots, \mathbf{e}_{(i_k)}], \tag{1}$$

where $\mathbf{e}_{(i)} \in \{0, 1\}^N$ is the $i$-th standard basis vector. Note that this formulation does not require that the same tokens be selected across each layer.

## 3.3 SCORING FUNCTION

The problem of determining which indices $(i_1, \ldots, i_k)$ to retain still remains. We would like the subselection $\mathbf{S}$ to retain most of the context information given a fixed $k$. One possibility is to use a feedforward neural network to measure the importance of each token position:

$$\mathbf{s} = \text{FFN}_\theta \left( \mathbf{X}'_\eta \right) \tag{2}$$

where $\theta$ is the parameters of the FFN, $\mathbf{s} \in \mathbb{R}^N$ and $\mathbf{s}_i$ indicate the "importance score" of the $i$-th token and $\mathbf{X}'_\eta$ indicates the hidden states at the $\eta$-th layer. The indices $i_{1:k}$ can then be derived by taking the tokens with the top-$k$ scores. We can control the percent of the KV cache retained by scaling $k$ with the length of the context. For the experiments that follow we retain the same $i_{1:k}$ across all layers and take $\eta = 6$.

The above selection procedure is rendered non-differentiable by the top-$k$ operator. We may propagate gradients to the scorer by decaying the attention weights of tokens attending to $\tilde{\mathbf{X}}$ inversely proportionally to their computed importance scores. More precisely, let $\mathbf{z} \in \mathbb{R}^d$ represent the hidden state of a single token attending to $\tilde{\mathbf{X}}$ with unnormalized attention weights $\alpha$:

$$\alpha = \left( \mathbf{z} \mathbf{W}^{\text{Q}} \right) \left( \tilde{\mathbf{X}}^{(K)} \right)^\top, \tag{3}$$

we decay $\alpha$ to produce scorer-informed attention weights $\alpha'$:

$$\alpha' = \sigma(\mathbf{s}) \odot \alpha, \tag{4}$$

where $\odot$ denotes the element-wise (Hadamard) product and $\sigma$ the sigmoid function. We note that the above formulation is one of many possible scoring functions that can be used with KV-DISTILL, that could be learnable or parameter-free, and could potentially have layer-wise specificity. We leave the problem of exploring different scoring functions to future work.

## 3.4 TRAINABLE PARAMETERS

After performing sub-selection to determine important token indices, we pass the context $\mathbf{c}$ through a modified $\text{LM}_\theta$ that uses *conditional computation* to condense the context into $\tilde{\mathbf{X}}$. This allows for the representations of important tokens to be "packed" with information from unselected tokens, and is strictly more expressive than only subselection. We instantiate $\text{LM}_\theta$ with LoRA adaptors (Hu et al., 2022) to minimize the number of trainable parameters. More importantly, within $\text{LM}_\theta$, the subselected tokens are routed to trainable $\mathbf{W}^{\text{Q}}, \mathbf{W}^{\text{O}}$

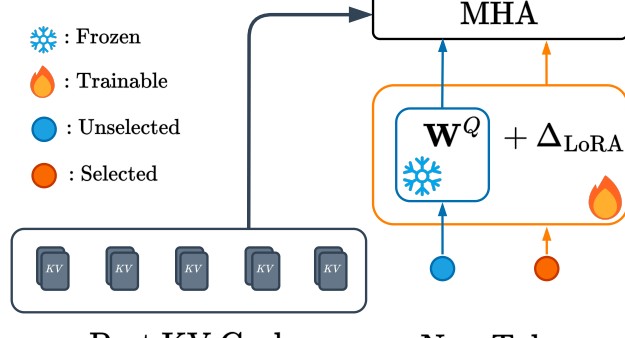

Figure 2: Selected tokens are routed to trainable, LoRA-adapted $\mathbf{W}^{\text{Q}}$ and $\mathbf{W}^{\text{O}}$ matrices ($\mathbf{W}^{\text{O}}$ is omitted in this figure), while other tokens are routed to frozen parameters.

matrices, where $\mathbf{W}^{Q}, \mathbf{W}^{O}$ are the query and output matrices of transformers, while discarded tokens are routed to the original (frozen) matrices, as shown in Figure 2. This has the effect of informing $\text{LM}_{\theta}$ as to which tokens are selected, allowing for specialized aggregation of the value representations for selected tokens. This method of informing $\text{LM}_{\theta}$ has minimal overhead (the LoRA matrices account for under 500MB of GPU memory for a 27B parameter model), and only a single set parameters must be maintained in memory.

We anticipate that other architectural forms could make KV-DISTILL effective. However, we find that applying conditional computation to inform $\text{LM}_{\theta}$ of selected tokens is important to the performance of KV-DISTILL . We find that some methods of informing the model of selected tokens, such as by adding a trainable embedding to these tokens, do not work well. The particular architecture chosen has the advantage of lower memory usage during training, and provides excellent performance. We leave the task of finding even more efficient architectures to future work.

### 3.5 OBJECTIVE FUNCTION

After generating compressed cache $\tilde{\mathbf{X}}$, we aim to match the output of LM when conditioned on $\tilde{\mathbf{X}}$ to the output of LM when conditioned on $\mathbf{X}$.

Previous compression methods (Ge et al., 2024; Qin & Van Durme, 2023) rely on the autoencoding objective to pretrain $\text{LM}_{\theta}$. However, given that LM predicts future tokens during inference, there is a discrepancy in pretraining and downstream usage, which could result in performance loss. Instead we propose matching the next-token probability distribution of tokens conditioned on $\mathbf{X}$ and $\tilde{\mathbf{X}}$.

Formally, consider a generative language model that predicts the next token $\mathbf{y}_t$ conditioned on the past tokens $\mathbf{y}_{<t}$ and a fixed context $\mathbf{c}$ that is represented by either $\mathbf{X}$ or $\tilde{\mathbf{X}}$. We would like to minimize the difference between their next-token distributions, i.e. $p\left(\mathbf{y}_t \mid \mathbf{y}_{<t}, \mathbf{X}\right)$ and $q_{\theta}\left(\mathbf{y}_t \mid \mathbf{y}_{<t}, \tilde{\mathbf{X}}\right)$.

Note that $q_{\theta}$ to indicate the distribution that conditions on the distilled KV cache $\tilde{\mathbf{X}}$. Also note that the only learnable parameters in this formulation arise from *encoding* $\tilde{\mathbf{X}}$; during auto-regressive generation we use the original frozen parameters of LM.

Given probability distributions $p, q_{\theta}$, we use the forward and reverse KL divergences to measure their similarity. With some simplified notations, we have

$$\mathcal{D}_{\text{KL}}(p\|q_{\theta}) = \mathbb{E}_{y \sim p(\cdot)}\left[\log\left(\frac{p(y)}{q_{\theta}(y)}\right)\right], \qquad \mathcal{D}_{\text{KL}}\left(q_{\theta}\|p\right) = \mathbb{E}_{y \sim q_{\theta}(\cdot)}\left[\log\left(\frac{q_{\theta}(y)}{p(y)}\right)\right] \quad (5)$$

Clearly the divergences are not symmetric, but can be made symmetric by summing the forward and reverse divergences:

$$\mathcal{L}(\theta) = \lambda \cdot \mathcal{D}_{\text{KL}}(p\|q_{\theta}) + (1 - \lambda) \cdot \mathcal{D}_{\text{KL}}(q_{\theta}\|p), \quad (6)$$

where a hyperparameter $\lambda$ controls the balance between forward and reverse KL divergence. We include both objectives in eq. (6) because of their strengths in mean- and mode- seeking behaviors.

Given both $p$ and $q_{\theta}$ are categorical distribution, both KL divergences in eq. (5) can be analytically solved. However, the L1-norm of the gradient of the reverse divergence dominates nearly everywhere. As such we propose scaling the forward and reverse terms by having $\lambda > 0.5$ in eq. (6).

Although the objective is no longer symmetric, the benefit of decaying reverse KL gradient norms by adding $\lambda$ is confirmed with the ablations in Section 5.6

## 4 EXPERIMENTS

To assess the efficacy of KV-DISTILL , we conduct experiments on LLAMA-2 7B, LLAMA-3 8B, MISTRAL 7B, GEMMA-2 9B and GEMMA-2 27B. In all cases we use the instruction-tuned model. We first pretrain all models on a large corpus to obtain general-purpose context compressors.

**Data** We curate a large instruction dataset from Self-Instruct, P3, LongAlpaca, and Super-Natural Instructions (Soboleva et al., 2023; Wang et al., 2022a; Sanh et al., 2021; Chen et al., 2023; Wang et al., 2022b). Training instances are split into (Context, Instruction, Answer) triples. We compress the context, leave the instruction uncompressed, and apply Equation 6 to tokens in the answer.

In cases where the context is sufficiently long (more than 1536 tokens), we pad to a multiple of 1536 and fold the context to a batch of $N \times 1536$ instances, compress the resulting KV cache, and then unfold the cache. Empirically, we observe little performance degradation when applying this technique during pretraining, while allowing the model to see longer examples during training. We also always leave the first few ($< 10$) tokens of the context uncompressed, as we find that retaining them improves performance; this is not a new observation, see Han et al. (2024).

**Training** We use rank-stabilized LoRA on the $Q, K, V, O$ matrices with $r = 128$ to train $\text{LM}_\theta$ (Hu et al., 2022; Kalajdzievski, 2023). Note that the $K, V$ are trainable for all tokens, not just selected tokens. The behavior of the $Q, O$ adapters is discussed in Section 3.4. Optimization is done using Deepspeed Stage 2, and the AdamW optimizer (Rasley et al., 2020). During pretraining, we sample KV retention fractions between $0.1 - 80\%$. As such, all KV-DISTILL models support arbitrary retention rates. See Appendix A for further training details.

Table 1: Dataset Statistics

| Dataset | Average | Max |
|---------|---------|-----|
| SQuAD | 225 | 1k |
| QuALITY | 6k | 9k |
| SQuALITY | 7k | 11k |
| GovReport | 10k | 71k |

**Evaluation** In all the evaluations that follow, we have a natural (Context, Question) pairing. In all cases we compress the context, while leaving the question uncompressed. All evaluations are performed with greedy decoding. Summary statistics regarding the context length of each dataset are provided in Table 1.

---

(a) 100% KV retention

The story follows Willard, a space explorer who is left alone after his friend and space mate, Dobbin, dies. Willard is haunted by the memory of Dobbin's final words, "I saw the Ghost Ship," and becomes obsessed with finding the ship. He spends years searching for it, and eventually, he sees a strange, ghostly ship that seems to be heading towards him. As he gets closer, he realizes that it's not just a hallucination, but a real ship from Earth that's been traveling through space for centuries. In the end, Willard is taken aboard the ship, but learns that the ship and its crew are mere shells of their former selves, and that they are doomed to remain in space forever. Ultimately, Willard is forced to confront the reality that he'll never return to Earth.

(b) 2.5% KV retention

The story follows the journey of Willard, a space traveler who has been alone in space for years after the death of his friend and partner, Dobbin. Willard is struggling to come to terms with his isolation and the loss of his friend, and is haunted by the legend of the Ghost Ship, a mythical vessel that is said to roam the solar system, never to be seen again. One day, Willard spots a strange, rocket-shaped object approaching him, and as it draws closer, he realizes it is the ==Mary Lou==. The ship is now a mere shadow of its former self, and Willard is invited to board it, where he is greeted by a rescue team who have been sent to retrieve him. As he adjusts to life on the rescue ship, Willard comes to terms with his past and the reality of his situation, and ultimately accepts that he will never return to Earth. The story concludes with Willard's poignant realization that he and the crew of the rescue ship are the Ghosts, doomed to roam the solar system forever.

(c) .25% KV retention

The story follows the journey of Willard, a space explorer who has been traveling through the galaxy for years, searching for a ==new home for humanity==. He has lost his friend and companion, Dobbin, and is struggling to come to terms with his own mortality. One day, he discovers a faint light in the distance, which he believes to be a sign of life. As he approaches the light, he realizes it is a ghost ship, a ship that has been drifting through space for ==millions== of years, its crew and passengers ==long dead==. The ship is a relic of a bygone era, and Willard is forced to confront the reality that he will never be able to return to Earth, as the ship would pass through it, making it impossible to survive. In the end, Willard comes to accept his fate, realizing that he and the crew of the ghost ship are doomed to drift through space forever, a reminder of the transience of human existence.

---

Figure 3: LLAMA-3 was tasked with summarizing a 6k token short story at low KV retention rates. Inaccuracies in the summary are highlighted yellow, and were determined by hand.

# 5 RESULTS

## 5.1 QUALITATIVE ANALYSIS

In Figure 3 we present representative summaries of the short story "Galactic Ghost". LLAMA-3 was prompted to summarize the short story in a few sentences. As the KV retention ratio decreases, we observe an increase in statements that are unfaithful to the original context. Particularly, at extremely high compression ratios, the model begins to hallucinate details, but impressively still maintains understanding of the general plot of the story. Note the varying degrees of *semantic compression* in summaries generated under high compression ratios; qualitatively, we observe that models with severely compressed contexts generally understand the text, but resort to making vague statements.

## 5.2 EXTRACTIVE QUESTION ANSWERING

**Motivation** SQuAD is an extractive question-answering task. We hypothesize that tasks that are extractive in nature would suffer the largest performance loss under context compression. As such, we choose to use performance on SQuAD as a proxy for general-purpose compressive ability of a model. In all the following experiments, we choose the pretraining checkpoint with the best SQuAD performance for further experimentation. To assess accuracy we generate an answer conditioned on the compressed context, checking whether the generated response is contained in the ground-truth answer.

Table 2: Accuracy on SQuAD at selected KV retention ratios. * indicates the model was not trained to convergence due to computational limitations

| Model | Method | KV retention | Accuracy |
|---|---|---|---|
| LLAMA-3 | BASE | 100% | $87.6 \pm .6\%$ |
| | KV-DISTILL | 25% | $86.6 \pm .7\%$ |
| | KV-DISTILL | 20% | $86.0 \pm .7\%$ |
| | $H_2O$ | 25% | $84.0 \pm .7\%$ |
| | $H_2O$ | 20% | $83.0 \pm .7\%$ |
| | DODO | 20% | $73.3 \pm .8\%$ |
| LLAMA-2 7B | BASE | 100% | $82.5 \pm .7\%$ |
| | KV-DISTILL | 25% | $79.1 \pm .8\%$ |
| | KV-DISTILL | 20% | $77.6 \pm .8\%$ |
| | $H_2O$ | 25% | $77.9 \pm .7\%$ |
| | $H_2O$ | 20% | $76.7 \pm .7\%$ |
| | ICAE | 57% | $75.0 \pm .8\%$ |
| GEMMA 9B | BASE | 100% | $85.15 \pm .7\%$ |
| | KV-DISTILL | 25% | $84.55 \pm .7\%$ |
| | KV-DISTILL | 20% | $83.1 \pm .7\%$ |
| GEMMA 27B | BASE | 100% | $85.3 \pm .8\%$ |
| | KV-DISTILL [*] | 25% | $83.1 \pm 1\%$ |
| | KV-DISTILL [*] | 20% | $82.2 \pm 1\%$ |
| MISTRAL 7B | BASE | 100% | $87.1 \pm .6\%$ |
| | KV-DISTILL | 25% | $84.1 \pm .7\%$ |
| | KV-DISTILL | 20% | $82.5 \pm .7\%$ |

**Results** Table 2 contains SQuAD accuracy results. We see that in all cases, KV-DISTILL models perform within a few percentage points of base models, even under a "worst-case" task. Furthermore, KV-DISTILL models significantly outperform prior trainable methods, even when retaining less of the KV cache. KV-DISTILL models also enjoy an improvement over $H_2O$ models at similar compres-

sion ratios, demonstrating the ability of the pretraining objective to "pack" the KV representations of important tokens. In practice, we find that performance on SQuAD increases monotonically with KV retention fraction.

When retaining under 20% of KV, we (informally) observe rapid declines in performance across all methods, indicating the difficulty of the task under high context compression. Lastly, we note that initial pretraining hyperparameters for all models were set based on initial experimentation with LLAMA-3 and SQuAD; as such, we anticipate that performance of most models can be improved with focused hyperparameter tuning during the pre-training process.

## 5.3  LONG CONTEXT QUESTION ANSWERING

**Motivation**  QuALITY is a long document multiple-choice question answering dataset that assesses reading comprehension. We use QuALITY to assess the decision making capabilities of models equipped with distilled contexts. To assess QuALITY accuracy, we use the same evaluation procedure used by LLAMA-3 (AI@Meta, 2024).

**Results** Figure 4 shows the experiment results on QuALITY, with data points at the following retention rates highlighted: $\{100\%, 25\%, 20\%, 10\%, 5\%, 1\%, 0.1\%\}$. We observe that KV-DISTILL performs similarly to the uncompressed cache, with only minor losses in performance at 10x compression. Although not included in Figure 4, 0% cache retention results in accuracy of 32.4%, 25.8%, and 24.4% for the LLAMA-3, MISTRAL, and GEMMA-2 models respectively, demonstrating the neccessity of the context for the task. Impressively, we see significant improvements over the random accuracy even when distilling to as few as 7 tokens from a 7k input passage; for example, on LLAMA-3 we observe only a 20% drop in accuracy despite eliminating 99.9% of the context.

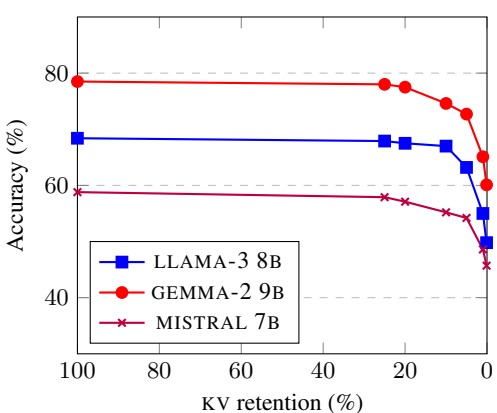

Figure 4: Accuracy on QuALITY

## 5.4  LONG CONTEXT ABSTRACTIVE SUMMARIZATION

**Motivation** SQuALITY is a question-focused summarization dataset based on the same collection of long documents as the QuALITY benchmark. We use it to evaluate the abstractive summarization capabilities of models trained with distilled contexts. We compute the Rouge-L scores (Lin, 2004) between the generated summaries and ground-truth answers, following the same evaluation protocol used by LLAMA-3 (AI@Meta, 2024).

**Result** Figure 5 show Rouge-L performance on SQuALITY (see Appendix B for results on Rouge-1 and Rouge-2). We observe that KV-DISTILL models perform as well or better than uncompressed models when retaining more that 20% of the KV cache. When retaining under 20%, we

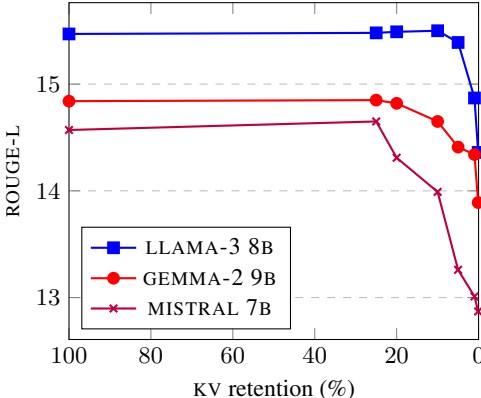

Figure 5: Rouge-L on SQuALITY

observe different performance falloff behaviors for different models; in particular, we observe that textscLlama-3 and GEMMA-2 have stable performance until 100x compression, at which point performance dips drastically. This difference in the behavior of the compression-performance trade-off could be attributed to the larger vocabulary sizes of LLAMA-3 and GEMMA-2, which allows the KL-loss to capture more fine-grained features of the output distribution during pretraining.These

results, and demonstrate that KV-DISTILL can support very high compression ratios with minimal performance penalty on abstractive tasks.

## 5.5 FINETUNED LONG CONTEXT SUMMARIZATION

**Motivation** GovReport is a long document summarization dataset that consists (Report, Summary) pairs written by government research agencies. In contrast to the evaluations on QuALITY and SQuALITY (which are performed in a zero-shot fashion using the best pretraining checkpoint), we perform additional training with Equation 6 on the Gov-Report training set before evaluation. As with SQuALITY, we use GovReport to assess the abstractive summarization ability of models equipped with KV-DISTILL .

Table 3: Experiment results on GovReport

| KV retention | $H_2O$ | Zero Shot | Finetune |
|---|---|---|---|
| 100% | 23.7 | 23.7 | 23.7 |
| 20% | 22.8 | 22.3 | **23.5** |
| 10% | 22.4 | 21.8 | **23.3** |
| 5% | 21.9 | 21.1 | **23.2** |

**Results** Table 3 shows results for GovReport for $H_2O$ , KV-DISTILL prior to finetuning (zero-shot) and KV-DISTILL after finetuning. We observe that KV-DISTILL and $H_2O$ perform close to each other on this evaluation in the zero-shot setting. However, upon finetuning, we observe a practical improvement in performance with KV-DISTILL , with little degradation from uncompressed performance across all compression rates. In particular, we note the improvement in performance is greater at more severe compression ratios, confirming the utility of KV-DISTILL in supporting ultra-high compression ratios.

## 5.6 PRETRAINING OBJECTIVE ABLATIONS

Lastly, we assess the necessity of both the forward and reverse terms in the loss by evaluating SQuAD performance on multiple different pre-training losses with varying $\lambda$ values in Equation 6. The results are given in Table 4. We observe that using either the pure forward or reverse divergences performs markedly worse than using a mixture of both. Furthermore, using solely the auto-encode + cross-entropy (used in ICAE and DODO), performs significantly worse than Equation 6, demonstrating the significant benefits that the weighted distillation objective provide.

Table 4: Effect of Pretraining Loss on LLAMA-3 SQuAD performance

| Loss | SQuAD Accuracy (%) |
|---|---|
| $\lambda = 1$ | 83.4% |
| $\lambda = 0.6$ | 86.0% |
| $\lambda = 0.4$ | 85.3% |
| $\lambda = 0$ | 82.7% |
| AE + LM | 79.1% |

## 6 DISCUSSION AND CONCLUSION

In this paper, we develop a method to reduce the memory requirements of long-context conditioned LM generation. Our method sub-selects tokens from the KV cache, and applies a token-level KL-type loss between the output of the LM when conditioned on sub-selected tokens and when conditioned on the uncompressed cache. We evaluate our method on long-context extractive and abstractive tasks, and demonstrate improved performance over competing compression methods. We further demonstrate that continued training on domain-specific data can allow for use of compression ratios as high as 20x with negligible losses in performance.

We plan to further explore the effect that different token sub-selection mechanisms may have on our method, including applying the non-trainable layer-wise selection mechanism described in $H_2O$ . We anticipate such a selection mechanism could yield a non-negligible improvement to the perfor-

mance of KV-DISTILL . Another interesting avenue is applying KV-DISTILL to multi-modal models, where some modalities such as vision may allow for very high compression ratios.

As part of this work we release distilled checkpoints across various model language families. These artifacts allow efficient text generation conditioned on significantly larger inputs than before, with much lower memory burden, and support compression ratios as high as 1000x. We anticipate these artifacts will be of great practical benefit to practitioners, enabling exciting new applications and research directions in language processing.

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

## A  TRAINING & EVALUATION DETAILS

We train all KV-DISTILL models with the following parameters at bf16 precision on 8 NVIDIA A100s. Please see Table 5 for further details.

Table 5: Hyperparameters for training

| Hyperparameter | Value |
|---|---|
| Optimizer | AdamW |
| Learning Rate | 5e-5 |
| Batch Size | 32 |
| LoRA Rank | 128 |
| $\lambda$ | 0.6 |
| $\eta$ | 6 |

## B  ADDITIONAL TABLES AND FIGURES

The QuALITY, SQuALITY, GovReport, and SQuAD evaluations are performed on the test set, if public, else results are reported on the development set. To measure SQuAD accuracy, we generate up to 128 tokens, normalize the output by stripping punctuation, and check if the correct answer is contained in the generated answer. For SQuALITY and QuALITY, we follow the evaluation procedure of AI@Meta (2024). For GovReport, we prompt the model to summarize the report, and then greedily generate 630 tokens.

Table 6: Tabulated SQuALITY Results

| Model | KV retention | Rouge 1 | Rouge 2 | Rouge-L |
|---|---|---|---|---|
| LLAMA-3 | 100% | 24.80 | 6.96 | 15.47 |
| | 25% | 24.95 | 6.75 | 15.48 |
| | 20% | 24.92 | 6.81 | 15.49 |
| | 10% | 24.79 | 6.61 | 15.50 |
| | 5% | 24.67 | 6.59 | 15.39 |
| | 1% | 23.89 | 6.04 | 14.87 |
| | .1% | 22.80 | 5.31 | 14.36 |
| GEMMA-2 9B | 100% | 23.02 | 5.91 | 14.84 |
| | 25% | 23.29 | 5.70 | 14.85 |
| | 20% | 23.05 | 5.76 | 14.82 |
| | 10% | 22.79 | 5.65 | 14.65 |
| | 5% | 22.94 | 5.51 | 14.41 |
| | 1% | 22.44 | 5.18 | 14.34 |
| | .1% | 21.67 | 4.70 | 13.89 |
| MISTRAL 7B | 100% | 22.43 | 5.86 | 14.57 |
| | 25% | 22.94 | 5.70 | 14.65 |
| | 20% | 22.50 | 5.69 | 14.31 |
| | 10% | 22.48 | 5.40 | 13.99 |
| | 5% | 21.81 | 5.41 | 13.26 |
| | 1% | 21.51 | 4.99 | 13.01 |
| | .1% | 20.86 | 4.94 | 12.87 |

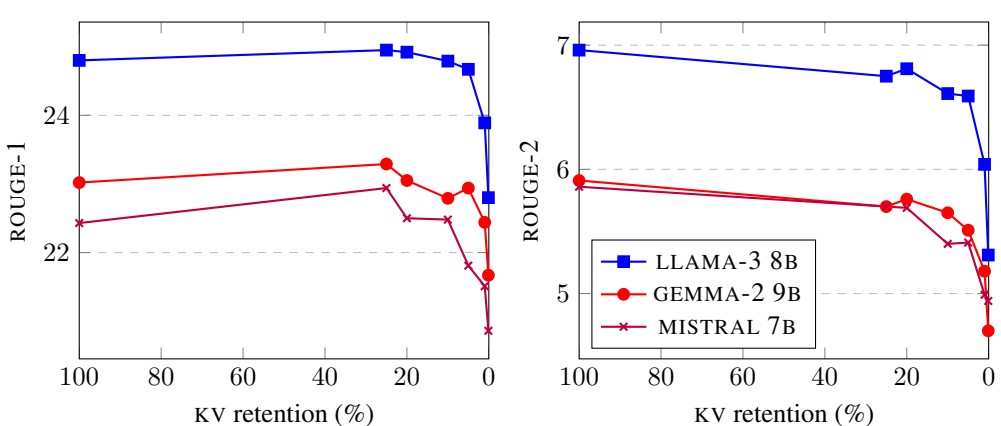

Figure 6: SQuALITY Results: ROUGE-1 & ROUGE-2

