# OpenReview forum: "KV-Distill: Nearly Lossless Context Compression for Transformers"
_ICLR.cc/2025/Conference — ICLR 2025 Conference Withdrawn Submission_

### Official Review · Reviewer_fWiV · 2024-10-28

**Soundness:** 2
**Presentation:** 3
**Contribution:** 2
**Rating:** 3
**Confidence:** 3

**Summary:**

The paper introduces a new context compression method.  The method works by training a scoring function, which picks which token embeddings are to be retained, as well as a LoRA adapted LLM, which compresses information into the remaining embeddings.  The newly added parameters are trained with a distillation objective with respect to the next token distribution of the uncompressed LM.  The method is evaluated on SQuAD as well as 3 long context tasks, and perform favorably to the baselines (mostly H20) in most cases.

**Strengths:**

* Good coverage of related work
* Great that the authors tried the method across multiple model families
* The method is demonstrated to work favorably to baselines in most cases

**Weaknesses:**

* Context compression by distillation is not a new idea as far as I know, and is the basis of earlier prompt compression and GIST token works.
* The presentation is not all that clear (see questions below).  The method could be better described, as well as why certain choices were made.
* While the relevant works section is good, I think the comparison with those approaches are not detailed enough.  Especially for H2O, I would have liked to see more detail, since this is the main baseline.
* It's also worth discussing the different settings for context compression.  For instance in one setting, we may assume a fixed context which is going to be reused, and the goal is to compress only this context.  Some settings might allow training, other might not.  Some use cases might allow the LM to be trained etc. etc.  The description of the use case, problem setting and motivation is not as crisp as it could be.
* Gains are not entirely consistent.  For instance on GovReport, H2O outperforms the proposed approach.  (Why is this?)

**Questions:**

* During training, why do discarded tokens need to be routed differently?  Isn't the scoring function meant to gate them anyway?
* Why do models other than llama3 (in particular llama2) degrade a lot more?
* It's a bit dubious that performance varies so wildly with the alpha parameter (the KL divergence objective).  Can other works/baselines be made to work better with similar optimizations?
* Can you report other ablations that you've tried? (e.g. you refer to one on line 225)
* How did you pick layer 6 for the scoring function?  Is it easy to allow different tokens for different layers?  Does it improve performance?
* Is it meaningful to compare to GIST tokens in their setting?

---

### Official Review · Reviewer_caXc · 2024-10-29

**Soundness:** 4
**Presentation:** 3
**Contribution:** 4
**Rating:** 6
**Confidence:** 5

**Summary:**

This article proposes a KV-Distill method to conduct general-purpose KV-Cache distillation on Large Language Models (LLMs). Specifically, KV-Distill trains a token "importance" scorer (a FFN module) with parameter-efficient LoRA modules during a pre-training phase and choose the Top-k scored tokens during inference for KV-Cache retention. Experiment results on multiple extractive and summarization tasks demonstrate that KV-Distill acquires general-purpose instruction-following KV-Cache distillers and outperforms the previous competing methods.

**Strengths:**

1. Originality: KV-Distill firstly distills the KV-Cache in LLMs across general-purpose tasks and under flexible compression rates.
2. Strong experiment results: While maintaining outstanding versatility, KV-Distill performs on-par with or slightly above the previous state-of-the-art methods (H$_2$O, ICAE, etc.). Moreover, KV-Distill performs consistently well on a series of downstream tasks.

**Weaknesses:**

1. Novelty: KV-Distill proposes to conduct KV-Cache eviction through unimportant token pruning, and proposes to insert a router-like structure (an FFN module) to conduct token selection. Currently, the idea of token eviction in the field of KV-Cache compression has motivated a number of methods [1,2]. Both methods [1,2] and KV-Distill propose to get rid of "unimportant tokens" through different importance measuring metrics.
2. Writing: This article lacks some essential details for better reproducibility. The authors should give a detailed description on how to train/deploy KV-Distill for an off-the-shelf model.
3. Applicability: The cost (i.e., GPU Memory cost, GPU time cost, total training parameters) for pre-training with the KV-Distill method is missing from the article. Additionally, as a compression method, the inference time speedup (such as tokens per second, or any other metric that reflects the throughput of LLMs) should be reported and compared to other methods.

[1] Keep the Cost Down: A Review on Methods to Optimize LLM' s KV-Cache Consumption

[2] NACL: A General and Effective KV Cache Eviction Framework for LLMs at Inference Time

**Questions:**

Things that WILL affect my overall rating towards this article:

1. For weakness 1: The authors should give a thorough comparison between the proposes KV-Distill and [1,2], and demonstrate the differences between the methods. Since [2] is an inference-time method, advantages of KV-Distill over [2] should be elaborated. Experiment comparisons with [1,2] are not necessary considering their publishing dates.
2. For weakness 2: The authors should include algorithmic pipelines or pseudo-code for training or inference with KV-Distill.
3. For weakness 3: Please consider including results/details mentioned in Weakness 3.

Things that WILL NOT affect my overall rating towards this article:

1. KV-Distill actually adopts the Top-p routing methods which resembles that in Mixture-of-Experts (MoE). What are other methods beside routing with an FFN (Line 178-179 states that FFN is "one possibility")? Additionally, the authors can consider comparing different routing/pruning strategies.

[1] Keep the Cost Down: A Review on Methods to Optimize LLM' s KV-Cache Consumption

[2] NACL: A General and Effective KV Cache Eviction Framework for LLMs at Inference Time

---

### Official Review · Reviewer_E8Tv · 2024-10-29

**Soundness:** 2
**Presentation:** 2
**Contribution:** 2
**Rating:** 3
**Confidence:** 4

**Summary:**

This paper introduces KV-Distill, a framework for nearly lossless compression of key-value (KV) caches in Transformer models, aimed at reducing GPU memory usage when handling long contexts. KV-Distill selectively retains essential tokens in the KV cache and applies a KL divergence-based loss between compressed and uncompressed representations, preserving model performance even at high compression rates (up to 1000x). The framework leverages LoRA adapters for efficient parameterization and maintains minimal performance degradation across various tasks, including long-context question answering and summarization.

**Strengths:**

Introducing the distillation method into KV cache compression can improve effectiveness.

**Weaknesses:**

1.	No comparison with SnapKV[1].

2.	Most of the compared methods are training-free, which is not entirely fair.

3.	Instruction-following ability is mentioned in the abstract, but there are no related experiments in the results section.

4.	There is no comparison on the complete LongBench[2] benchmark.

5.	Needle-In-A-Haystack is not included in the comparisons.

6.	No information is provided about the training cost for distillation.

7.	Lines 071-076 mention certain limitations of previous work, such as requiring extra training or using BLEU for evaluation, but it’s not clear how your work addresses these issues.

8.	Table 2 should include values similar to the zero-shot column in Table 3 to better highlight the advantages brought by distillation.

> Reference
> [1] Li, Yuhong, et al. "Snapkv: Llm knows what you are looking for before generation." arXiv preprint arXiv:2404.14469 (2024).
> [2] Bai, Yushi, et al. "Longbench: A bilingual, multitask benchmark for long context understanding." arXiv preprint arXiv:2308.14508 (2023).

**Questions:**

Please check the weaknesses section.

---

### Note · Authors · 2024-11-25

**Comment:**

We have taken time to carefully review the reviewers’ feedback and thank them for the time and effort they put in, but will we be withdrawing the submission from the conference.  We will incorporate the feedback to strengthen the work going forward.

**Withdrawal Confirmation:**

I have read and agree with the venue's withdrawal policy on behalf of myself and my co-authors.